# Electrowetting on Dielectric (EWOD) Device with Dimple Structures for Highly Accurate Droplet Manipulation

**Katsuo Mogi [1,2,\*], Shungo Adachi [1] , Naoki Takada [2] , Tomoya Inoue [2] and Tohru Natsume [1]**

[1] Molecular Profiling Research Center for Drug Discovery, National Institute of Advanced Industrial Science and Technology (AIST), Tokyo 135-0064, Japan; s.adachi@aist.go.jp (S.A.); t-natsume@aist.go.jp (T.N.)

[2] Research Center for Ubiquitous MEMS and Micro Engineering, National Institute of Advanced Industrial Science and Technology (AIST), Ibaraki 305-8564, Japan; naoki-takada@aist.go.jp (N.T.); inoue-tomoya@aist.go.jp (T.I.)

\* Correspondence: mogi.k@aist.go.jp; Tel.: +81-3-3599-8251

**Abstract:** Digital microfluidics based on electrowetting on dielectric (EWOD) devices has potential as a fundamental technology for the accurate preparation of dangerous reagents, the high-speed dispensing of rapidly deteriorating reagents, and the fine adjustment of expensive reagents, such as the preparation of for positron emission tomography (PET). To allow single substrate type EWODs to be practically used in an automatic operation system, we developed a dimple structure as a key technique for a highly accurate droplet manipulation method. The three-dimensional shape of the dimple structure is embossed onto a disposable thin sheet. In this study, we confirmed that the dimple structure can suppress unintended droplet motion caused by unidentified factors. In addition, the stability of the droplets on the dimple structures was evaluated using a sliding experiment. On a flat substrate, the success rate of a droplet motion was lower than 70.8%, but on the dimple structure, the droplets were able to be moved along the dimple structures correctly without unintended motion caused by several environmental conditions. These results indicated that the dimple structure increased the controllability of the droplets. Hence, the dimple structure will contribute to the practical application of digital microfluidics based on single substrate type EWODs.

**Keywords:** digital microfluidics; electrowetting on dielectric; EWOD; dimple structure; sliding method; droplet

## 1. Introduction

Various fields—e.g., medicine, drug discovery [1], industrial chemistry [2,3], and basic chemistry [4]—are supported by basic wet-lab experiments based on handling micro reagents with pipettes. In recent years, statistical analyses of large amounts of data using computer technology have become possible [5–7]. With the progress in technology, wet-lab experiment results are becoming proportionally required in large quantities [8,9]. Hence, an enormous burden is being placed on researchers who perform wet-lab experiments with simple pipetting operations. Consequently, the abilities of excellent research personnel are being wasted and human error caused by the decline in the researchers' attention are becoming major problems. Additionally, simple operations should be automated to enable the high reproducibility of reagent adjustments, because personal differences in manual dispensing operations can greatly affect the results of the adjustments. Hence, there are some groups that promote mechanical automation [10], but most other groups have difficulty installing expensive, large equipment.

On the other hand, research related to microfluidics using liquid-handling devices [11], e.g., microfluidic devices [12–16] and electrowetting on dielectric (EWOD) devices [17,18], has been actively carried out with the advances in microfabrication technology in recent years. These techniques, which deal with a small amount of liquid using microchannels and electrodes, are attracting attention as automated techniques for tedious pipetting operations. In particular, the EWOD device is expected to become a general purpose device because it can be used to handle various liquids by simply changing the signal patterns [17,18], in contrast to microfluidic devices, where the processing content is uniquely determined by the restrictions of the channel shape. In addition, the EWOD device is compatible with computer technology and, therefore, appropriate for practical use, since the motion of a microdroplet can be directly controlled by a digital signal for electrical switching [19,20]. Especially single substrate type EWODs have a high compatibility with various pipetting processes, since we can directly approach droplets on the substrate with a pipette.

However, the weak electrostatic force of the EWOD for droplet manipulation may cause malfunctions due to a tilted or distorted substrate, a minute scratch, dirt, etc. Therefore, highly accurate droplet manipulation is strongly desired, along with high reproducibility independent of experimental conditions with various possible disturbances.

We devised an open style EWOD device with a dimple structure embossed onto a thin disposable sheet to prevent droplets from overshooting and sliding in unintended directions. Owing to the dimple structure, droplets can be accurately and reliably transferred to the intended location. In this study, the effects of the dimple structure on droplets were verified using a model based on a sliding experiment, showing that the dimple structure realized highly accurate droplet manipulation.

## 2. Methods

### 2.1. EWOD Device Fabrication

Dimples are used for the purpose of producing a small effect to slow down a droplet, but it is an important effect that helps with position control. Although there are many parameters, such as curvature and depth, that optimize the dimple shape, the range of operable droplet sizes mainly depends on the size of the electrode tile, the gap between the electrodes, and the thickness of the insulating film. Hence, in order to introduce the beneficial effect of the dimple and the curved shape fabrication technique, which is difficult to achieve by conventional microfabrication in an easy-to-understand manner, we demonstrated a single condition as a representative example rather than the examination of a large amount of parameters.

The structure of the EWOD device in this study was a laminate of an insulation layer and an electrode layer, as shown in Figure 1. The end shape of all electrodes on the device was a jagged tile, 2 mm square, and dimple structures were embossed at the center of the tiles. To facilitate the embossing, a thin paper sheet (Circuit Paper, TAKEO Co., Ltd. Tokyo, Japan) was used as the electrode layer [21,22], instead of a solid material such as glass, epoxy resin, or acrylic resin. Thus, the sheet can be easily deformed into any shape by embossing with a low load.

The electrodes were patterned onto the paper sheet by an ink-jet printer (PX-S160T, EPSON, Suwa, Nagano, Japan), as shown in Figure 2A. Subsequently, a 10–15 μm-thick paraffin membrane was laminated as an insulation layer onto the paper sheet on which the electrode was printed (Figure 2B). Next, a 2 mm diameter sphere was pressed into the laminated sheet using a commercially available embossing device (CURIO, Silhouette America, Inc., Lindon, UT, USA) to form a dimple structure in the shape of a spherical cap (Figure 2C). The spherical-cap shape without any edges can avoid an unnecessary pinning effect of droplets. Finally, Fluorinert (FC-70, Sigma-Aldrich Co., St. Louis, MO, USA) of 2–3 μL as a water repellent was poured with a pipette onto the surface of the laminated sheet with emphasis on a high lubricating ability and ease of production (Figure 2D). A liquid film of less than 5 μm in thickness was formed on the surface. Although this thickness greatly affects the motion of the droplets in conventional EWODs, this effect could be ignored using our proposed method.

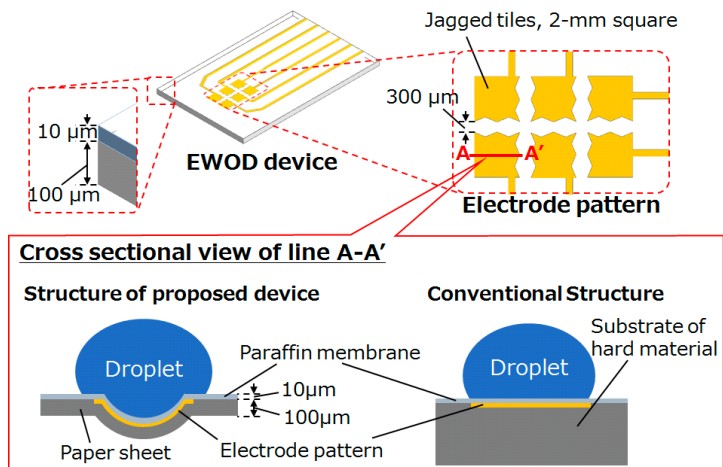

**Figure 1.** Structure of the electrowetting on dielectric (EWOD) device with a dimple structure. The device was composed of an insulation layer and an electrode layer. The terminal shape of the electrode was a jagged tile, 2 mm square, and the interval between the tiles was 300 μm. The electrode layer was a 100 μm paper sheet, instead of a substrate made of hard material. The paraffin membrane was 10 μm thick. A dimple structure with a spherical-cap shape was formed on each tile for droplet trapping.

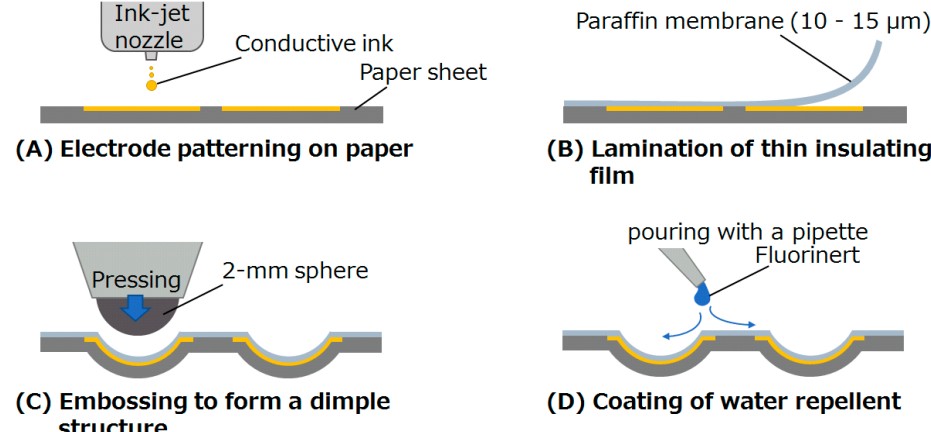

**Figure 2.** Fabrication process of the proposed the EWOD device. (**A**) The electrode pattern was printed on a paper sheet by ink-jet printing. (**B**) A paraffin membrane, as an insulation layer, was laminated onto the paper sheet. (**C**) Spherical dimples were formed on the laminated sheet by pressing with a 2 mm steel ball. (**D**) The surface of the laminated sheet was coated with Fluorinert as a water repellent.

*2.2. Effect Verification of the Dimple Structure*

To discuss the effect of the dimple structure, we needed to verify the stability of the droplet held in the dimple structure. The stability represented force that was necessary to move the droplet out of the dimple structure.

However, the material of the EWOD device with the dimple was the same as the EWOD device without the dimple, and the existing theory based on contact angles could not be applied to the EWOD force evaluation. Therefore, we proposed a new evaluation model that replaced the EWOD force acting on droplets with gravity. Although the EWOD force acting on the boundary of the droplet and solid surface was different from the gravity acting on the mass, we obtained one index of the dimple effect by the evaluation model which substituted the force by EWOD with gravity.

In order to demonstrate the stability of the droplet, the motion of the droplet on the EWOD device was modeled by observing the sliding motion of the droplet on an inclined surface, and the force per

unit length of the contact perimeter (adhesion energy) was calculated using Wolfram's experimental formula (1) [23,24].

$$K = \frac{mg \sin \alpha}{2\pi r}, \tag{1}$$

where $K$ is a constant approximating the adhesion energy, $m$ is the droplet mass, $g$ is the acceleration due to gravity, $\alpha$ is the critical tilting angle (sliding angle), and $r$ is the droplet contact radius. In this study, $r$ was considered to be the apparent radius without the dimple shape.

## 3. Experiments

### 3.1. Droplet Manipulation

In our experiment, we demonstrated highly accurate droplet manipulation using the EWOD device with dimple structures. Figure 3 shows the experimental setup for droplet manipulation. The six electrodes were connected to a voltage source (High Voltage DC-DC Boost Converter, 200–450 V, NoData, China) via six relay switches (G5V-1, Omron, Kyoto, Japan). The switching was controlled by a microcomputer (PIC16F628A) and a series of commands for the switching was issued using Python (Ver. Python 3.6.0, Python Software Foundation, Open source) in the control PC (ThinkPad X201s, Lenovo, Beijing, China). The voltage was set to 320 V and the switching interval was set to 1.0 s. The switching of the voltage allocation generated the electrostatic force to move the droplet, which was captured as MP4 data by a camera (EOS Kiss X8i, Canon Inc., Tokyo, Japan). The droplets used for verification were 10 and 20 µL of distilled water. To visualize the actual path of the droplet, we extracted images every second from the MP4 data and measured the median points of the droplets using ImageJ (ImageJ Ver. 1.45s, public domain, Open source).

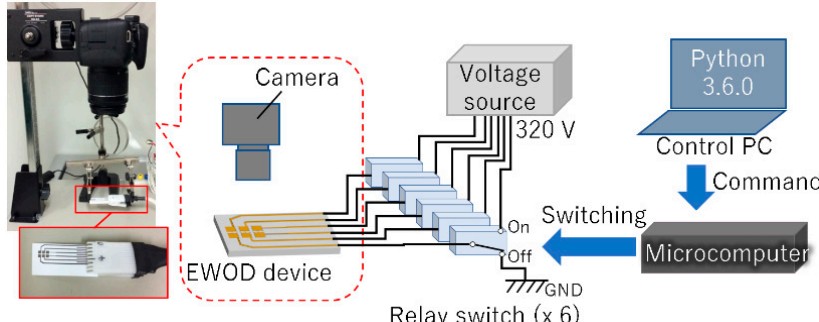

**Figure 3.** Photo and schematic of experimental setup for droplet manipulation by EWOD. All electrodes of the EWOD device were connected to each relay switch. The applied voltage for the EWOD was switched by a command from the control PC.

### 3.2. Sliding Experiment

To estimate the difference between the force required for the droplet to slide on the dimple structure and on a flat surface, the sliding angle was measured by the experimental setup shown in Figure 4. The droplets used for the verification were 5, 10, and 20 µL of distilled water, and the tilt stage was gradually tilted from 0 to 90 degrees to measure the sliding angle. The tilting test was performed five times for each volume, while taking videos with a camera (EOS Kiss X8i, Canon Inc., Tokyo, Japan). To calculate the force, we extracted an image from the video and measured the sliding angle using ImageJ (ImageJ Ver. 1.45s, public domain).

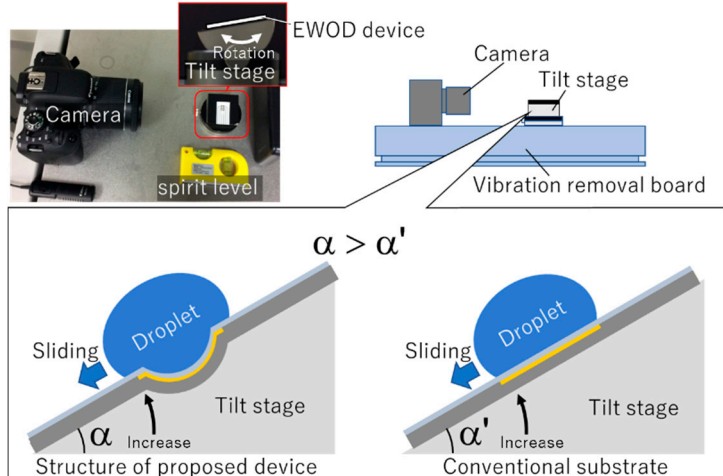

**Figure 4.** Photo and schematic of experimental setup for the sliding experiment. The EWOD device with and without the dimple structure was placed on the tilt stage. A cross-sectional view of the droplet motion was captured by a camera to measure the sliding angle $\alpha$ ($\alpha'$). The tilted angle of the stage was gradually increased from 0 to 90 degrees.

## 4. Results and Discussion

### 4.1. Droplet Manipulation

We fabricated a demonstration device with six electrodes, each with a 2 mm$^2$ jagged tile as the terminal shape; the interval between the tiles was 300 μm. As shown in Figure 5, the dimple structure was 47 μm in depth and the shape was a gently sloping spherical cap. The 2 mm$^2$ tiles corresponded to a 10–20 μL droplet manipulation. In fact, the lower limit of the operable droplet size was between 2 μL and 5 μL and the upper limit was between 20 μL and 25 μL, regardless of the presence or absence of dimples.

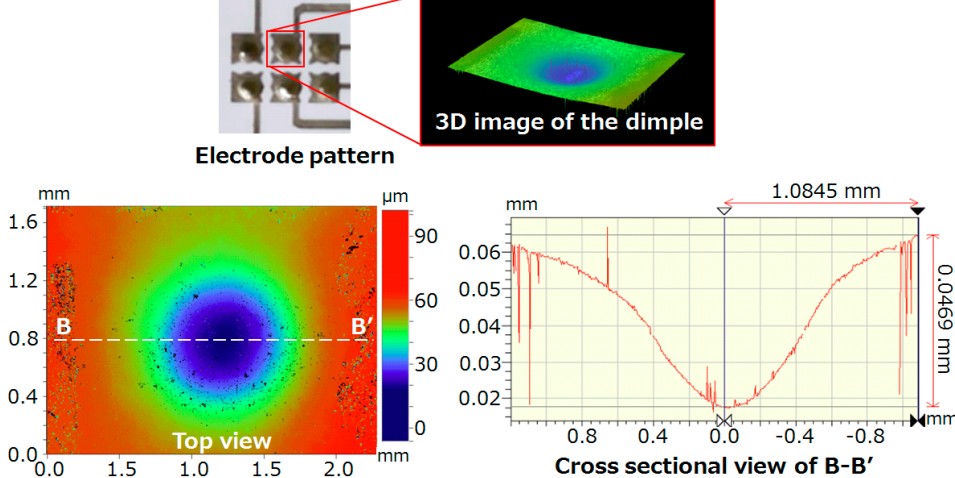

**Figure 5.** Three-dimensional image of the dimple structure. The dimple structure was 47 μm deep, with the shape of a spherical cap. The boundary between the dimple structure and the flat surface was gently sloping.

Figure 6 shows the paths of the median point of the droplets, traced at every second. On the device with a dimple structure, the droplet was stabilized on the dimple structure and moved accurately to the desired position, as shown in Figure 6A. All droplets moved correctly between tiles as commanded. This showed that the dimple structures enabled highly accurate manipulation of the

droplets, independent of minor conditional differences (Video S1). Additionally, even if the wiring paths and electrode tiles were fabricated on the same layer (with a lower cost), the droplets could be stabilized at the desired positions without being affected by the wiring path.

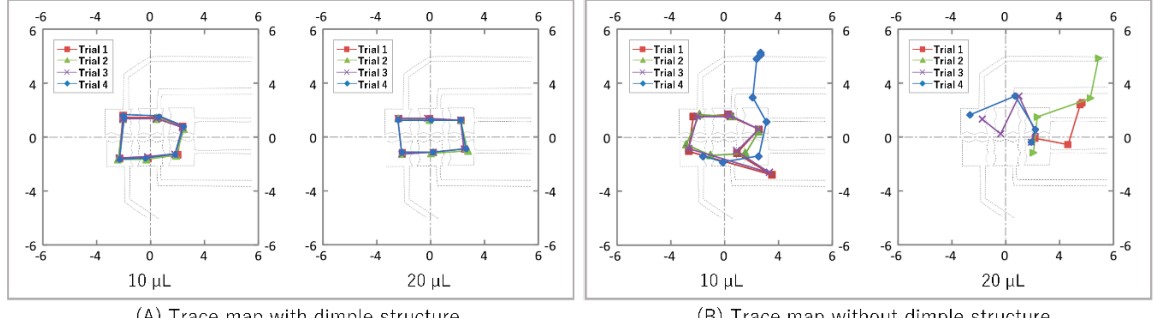

**Figure 6.** Droplet migration pathway on the EWOD device. (**A**) Trace map of the droplets on a surface with a dimple structure. (**B**) Trace map of the droplets on a surface without a dimple structure. The volumes of the droplets were 10 µL and 20 µL. The median point of the droplets was plotted every second. Each coordinate plane was 12 mm square, and the coordinate origin was at the center of the six tiles. The dotted lines represent the boundaries of the electrodes and each of the four trials is indicated in a different shape and color.

On the other hand, when using the device without the dimple structure, minor differences in the environmental conditions appeared to greatly affect the motion because of the weak electrostatic force of the EWOD, since the droplets often moved in an unintended direction, as shown in Figure 6B (Video S2). The percentage of droplets correctly moving between tiles as commanded was 70.8% for 10 µL droplets, and 16.7% for 20 µL droplets. In particular, in the case where the tile and the electrode path were designed on the same layer, the droplets often drifted along the path towards the voltage source without staying on the tile. Therefore, to avoid this drift phenomenon, according to conventional theory, the wiring path connecting the tile and the voltage source would need to be raised apart from the surface. However, a three-dimensional structure complicates the fabrication process, and the time and cost become enormous.

### 4.2. Sliding Experiment

Dimple structures in the proposed method were formed on the laminated sheet to hold and stabilize the droplets. It was difficult to discuss the effect of the dimple structure with a conventional method using the contact angle because the material composition of the sheet was the same as the flat surface and the contact angle was also the same. Hence, we measured the sliding angles to derive the adhesion energy of the surface and estimated the stability of the droplet on the dimple structure. Figure 7 shows the results of the sliding experiment in which devices with and without the dimple structure were compared. As shown in Figure 7A, the sliding angles decreased as the droplets increased, and the angles of the surface with the dimple structures were larger than those of the flat surface. From this sliding angle, we derived the adhesion energy of each surface. The result showed that the value of the surface with dimple structures was approximately 1.21 mJ/m$^2$ higher than that of the flat surface (Figure 7B). On the dimple structure, which had a large adhesion energy, it can be considered that the potential energy was relatively low (energetically stable), which means that the droplet was stable. Additionally, the small error bar, which represented the standard deviation of the five results, showed stable data with low variation. This low variation in the adhesion energy ensured reliable droplet motion, independent of differences in experimental conditions.

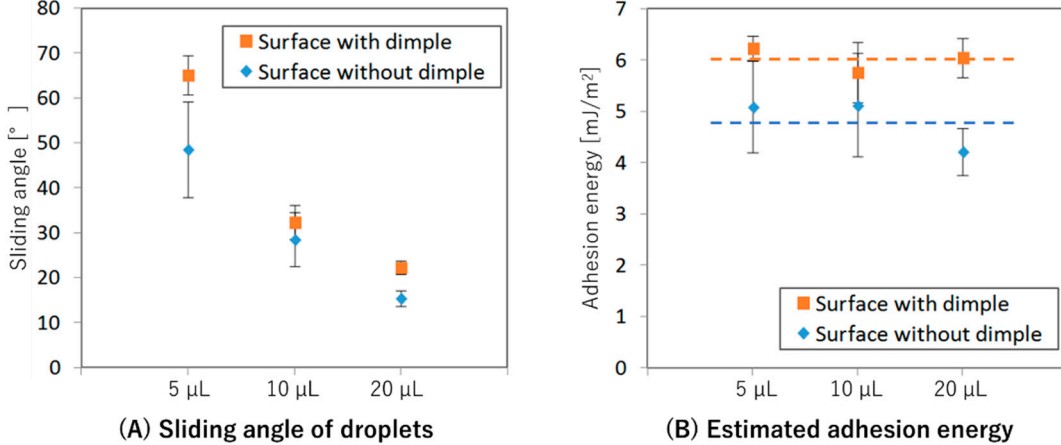

**Figure 7.** Results of sliding experiment. (**A**) Graph of the sliding angle for each condition. (**B**) Graph of adhesion energy calculated from the sliding angle. The volumes of the droplets were 5 μL, 10 μL, and 20 μL. The plotted points are the average of five data points, and the error bar shows the standard deviation. Dotted lines on the graph (**B**) represent average values of the three data of each surface.

## 5. Conclusions

In this research, we proposed a dimple structure to realize the accurate digital manipulation of droplets using EWOD devices. The three-dimensional shape of the dimple structure was successfully formed by simple embossing. On a flat substrate, the success rate of the droplet motion was lower than 70.8%, but on the dimple structure, the droplets were able to be moved along the dimple structures correctly without unintended motion caused by several environmental conditions. In addition, it was confirmed using a sliding experiment that the adsorption energy, which is an indicator of the droplet stability, increased by 25%. As a result, it was proved that the dimple structure increased the controllability of the droplets.

Stabilizing the droplet at the position of the dimple structure can reduce the manufacturing costs because the wiring path and the electrode tile can be formed on the same layer of a disposable sheet. In the near future, low-cost and highly accurate droplet manipulation may become possible by integrating EWOD devices with a dimple structure into basic wet-lab experiments. In other words, it is expected to achieve practical use as an automatic operation system that can greatly contribute to the biochemical industry.

**Supplementary Materials:** The following are available online at http://www.mdpi.com/2076-3417/9/12/2406/s1, Video S1: Droplet manipulation with dimple, Video S2: Droplet manipulation without dimple.

**Author Contributions:** Conceptualization, K.M., T.I. and T.N.; methodology, K.M., and N.T.; validation, K.M.; writing—original draft preparation, K.M.; writing—review and editing, K.M., and S.A.; funding acquisition, K.M. and T.N.

**Funding:** This work was partially supported by RIKEN-AIST joint research fund, the Ministry of Education, Science, Sports and Culture, Grant-in-Aid for young scientists (B) (No. 17K17715) and Leading Initiative for Excellent Young Researchers (LEADER) in FY 2016 of the Ministry of Education, Culture, Sports, Science and Technology, Japan.

**Acknowledgments:** We would like to thank the 4 University Nano Micro Fabrication Consortium in Kawasaki, Japan (http://www.nano-micro.sakura.ne.jp/home/) who provided open facilities and experimental equipment for this work. We would like to thank Editage (www.editage.jp) for English language editing.

**Conflicts of Interest:** The funders had no role in the design of the study; in the collection, analyses, or interpretation of data; in the writing of the manuscript, or in the decision to publish the results.

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
