# Peer review of "Electrowetting on Dielectric (EWOD) Device with Dimple Structures for Highly Accurate Droplet Manipulation"

_applsci, doi:10.3390/app9122406_

Round 1

Reviewer 1 Report

The authors presented an interesting strategy to stabilize the droplet on a one-plate EWOD platform by creating a dimple on the electrode to hold the droplet in position for more controlled droplet operation. The proposed approach is simple and effective, but not sufficiently characterized. Two major issues,

1.     The authors probably need to test dimples of difference sizes and depths and identify the limit of the system. For example, at what depth the droplet can no longer be moved out of the dimple.

2.     The authors attributed the droplet pinning to surface adhesion? Could potential energy play a roll, especially when the dimple becomes deeper?

Reviewer 2 Report

This paper is about stability of droplet manipulation in EWOD devices.This paper is well designed and written and is suitable for publication in Applied Sciences. However, it must be improved in writing and structure of the paper to enhance the interest of the readers. The paper can be published in Applied Sciences after suggested corrections.

Abstract: 

The abstract should be sharp and clear. Few suggestions are listed here. 

The word microfluidics should be used in the whole document. 

Lines 11-3, the sentence is very generic and should be simplified while more attracting towards EWOD devices. 

"Dimple structure has increased controllability" by what extent, any number or parameter to compare with. It is unclear at the moment or the claim is weak in the abstract.

Introduction: 

The introduction is logically written well. Few points

Line 29. What is wet-experiment?

Line 36-37. reference for some groups. the word some group is unclear.

Materials and Methods:

Material and Methods should come after Introduction, Then Results and Discussions and Then Conclusions. Experiment should come under Materials and Methods.

There is no materials sub-section present.

Results and Discussions:

Fig.6 can be added in results.

Conclusions:

Should be precise and also have some parameters to compare with other technologies. Passive voice should be used in the sentences such as we should be ignored.

Reviewer 3 Report

This manuscript described a novel EWOD substrate with dimple on each electrode to stablilize the droplet actuation repeatibility and avoid wiring electrode disturbance. The design, fabrication, and experimental data is elegant and directly prove the concepts. As a proof of concept of dimple EWOD, this manuscript is good in terms of demonstration. More application data, such as real biological sample, protein droplets, and droplets with different conductivity are recommended to further polish the manuscript. In addition, it would be very interesting if sandwich dimple structure can be test on the droplet actuation movement.

Paper material was used as substrate in this study. It is recommended if the author can try more traditional solid substrate with dimple structure fabrication compatible. In addition, hydrophobic dry coating would be more preferable in real application if author could demonstrate the data.

Some additional comments and typos are listed below :

In the second page, line 64-65, "On a flat surface without a dimple structure, the droplet...." Is the "without" a typo instead of "with"?

In the introduction part of page 1 on microfluidics, the authors might cite some manuscripts, which discussed in detailed about 3D polymer and microelectrode microfabrication integration on small volume sample processing and channel geometry.

Round 2

Reviewer 1 Report

The revised manuscript looks fine and ready for publication, although it could certainly be improved with more data on device characterization. I think the author should include some of the information in the response into the manuscript, e.g. why can't be dimples be made into other shapes?

Author Response

Dear Sir,

Thank you for giving us your precious time on this article.Your suggestion has greatly helped revise our article. Thank you again for reviewing this paper carefully.

Based on your comments, we have revised the paper as follows. If you find it different from your intention, I'm very sorry and please do not hesitate to let us know.

Revision is marked as purple in the revised manuscript.

Line64-71: Dimples are just for the purpose of producing a small effect to slow down the droplet, but it is an important effect that helps position control. Although there are many parameters such as curvature and depth to optimize the dimple shape, the range of operable droplet sizes is mainly depend on the size of the electrode tile, the gap between the electrodes, and the thickness of the insulating film. Hence, In order to introduce the dimple’s beneficial effect and the curved shape-fabrication technique that is difficult to make by conventional microfabrication in an easy-to-understand manner, We demonstrated single condition as a representative example rather than the examination for a large amount of parameters.

Line83-84:device (CURIO, Silhouette America, Inc.) to form a dimple structure in the shape of a spherical cap (Figure 24(C)). The spherical-cap shape without any edges can be avoid unnecessary pinning effect of droplets. Finally